# Social Touch: Its Mirror-like Responses and Implications in Neurological and Psychiatric Diseases

Laura Clara Grandi [1,*] and Stefania Bruni [2]

1    Department of Biotechnology and Biosciences, NeuroMI (Milan Center of Neuroscience),
     University of Milano-Bicocca, Piazza della Scienza 2, 20126 Milano, Italy
2    Centro Cardinal Ferrari, 43012 Fontanellato, Italy
*    Correspondence: laura.grandi@unimib.it

**Abstract:** What is the significance of a touch encoded by slow-conducted unmyelinated C-tactile (CT) fibers? It is the so-called affiliative touch, which has a fundamental social impact. In humans, it has been demonstrated that the affiliative valence of this kind of touch is encoded by a dedicated central network, not involved in the encoding of discriminative touch, namely, the "social brain". Moreover, CT-related touch has significant consequences on the human autonomic system, not present in the case of discriminative touch, which does not involve CT fibers as the modulation of vagal tone. In addition, CT-related touch provokes central effects as well. An interesting finding is that CT-related touch can elicit "mirror-like responses" since there is evidence that we would have the same perception of a caress regardless of whether it would be felt or seen and that the same brain areas would be activated. Information from CT afferents in the posterior insular cortex likely provides a basis for encoding observed caresses. We also explored the application of this kind of touch in unphysiological conditions and in premature newborns. In the present literature review, we aim to (1) examine the effects of CT-related touch at autonomic and central levels and (2) highlight CT-related touch and mirror networks, seeking to draw a line of connection between them. Finally, the review aims to give an overview of the involvement of the CT system in some neurologic and psychiatric diseases.

**Keywords:** social touch; CT fibers; mirror neurons; social brain; massage therapy





## 1. Introduction

Both humans and nonhuman primates exchange emotionally and socially relevant signals through touch [1]. Social touch helps create and maintain bonds among individuals of the same species and groups. For example, grooming is critical for building lasting and reassuring social bonds in nonhuman primates. About "grooming", it is well established that there is a social function behind this behavior, compared with the more obvious one of cleaning, which explains why individuals spend more time grooming others than is necessary. This is even more significant if we consider that during this period, the level of vigilance of both groomers and groomed individuals is lower than it should be, implying that they may not react immediately against predators in the event of an attack [2–4].

Similarly, in humans, touch has a crucial role in interactions with the external world: It is the first sense to develop during ontogeny, [5], and it plays an early and pivotal role in social interactions [6]. Indeed, decreased exposure to social touch during development, either due to its unavailability (e.g., as in the case of preterm infants placed in incubators or of infants of mothers with postpartum depression) or to atypical touch perception (e.g., as might be in autism) has serious consequences for subsequent brain and cognitive development [7], such as a reduction in grey matter in adults [8,9].

In the last decade, it has been demonstrated that C-tactile (CT) unmyelinated afferents contribute to pleasant touch and provide the neurobiological substrate for interpersonal

touch transmission [10,11]. These conclusions are supported by the data collected with different approaches, including brain imaging studies, studies carried out in animals, healthy human subjects, patients with a deficit in the CT system, analyses of autonomic and central effects elicited through CT activation, and behavioral studies. A growing body of literature supports the idea that the CT system belongs to the interoceptive system, which is known to be involved in encoding the emotional components of the touch. In humans, the "social brain" has been proposed as the neural network underlying the coding of CT-related touch. This circuit involves specific brain regions, including the posterior insular cortex, a critical cortical target for CT afferents. Interestingly, the same area seems involved in coding the observed caresses, but only if the touch is given at the optimal speed to activate the CT fibers [12]. However, investigating neural mechanisms for vicarious touch requires extending the study of the relevant neural network to include the putative mirror neuron system [7].

Considering the social role of the CT system, pleasant touch in psychiatric disorders characterized by a deficit in social behavior has also been explored. From the analysis of the literature, it can be inferred that there are a few disorders relevant to both mirror neurons and CT systems, such as autism spectrum disorder. Beyond this, a growing body of evidence suggests that a decreased activation of the mirror neuron system may be involved in the pathophysiology of psychiatric diseases characterized by social impairments, such as schizophrenia and psychopathy [13]. On the other hand, other psychiatric disorders mainly may involve the CT-fiber system [14] (e.g., anorexia nervosa, anxiety, and post-traumatic stress disorders).

Despite the excitement surrounding this topic, and its clinical relevance, it remains poorly defined and understood. Here, we first review seminal papers on CT fibers to highlight the effects on the autonomous and central nervous systems that enable CT fibers to play a role in coding the social and emotional aspects of the touch. We then discuss papers about (1) social touch, CT fibers, and mirror neurons, as well as (2) social touch, CT fibers, and psychiatric disorders. Overall, the analysis of the literature should drive future studies on the neural mechanisms underlying the computation of pleasant touch perception in the human and nonhuman primate brain and will ultimately promote the personalization of rehabilitative interventions in several clinical populations.

## 2. CT Fibers: The Discovery

Neurophysiological and neuroanatomical studies have demonstrated that discriminative and affiliative types of touch are encoded in two different ways. Traditionally, research on touch has mostly focused on the class of receptors responsible for transducing information about pressure/vibration, temperature, itch, and pain, i.e., the discriminative system. In comparison, the affective system has been particularly studied in recent decades.

CT fibers were first identified in cats in 1939, and they have been found in different species and body sites ever since (i.e., mice, guinea pigs, rats, pigs, and nonhuman primates) [15–19]. As they were identified in humans only in 1988, for a long time, it was believed that these fibers had disappeared in humans for an evolutionary reason. Subsequently, they were found in the infraorbital nerve, the hairy side of the arm, and the legs [20,21]. Therefore, the idea that CT fibers were absent in humans due to being unnecessary, was overruled and then replaced by research on the skin and the importance of emotional valence of touch, as well as by evidence that some disorders (e.g., depression) have deficits in this social system. We will deepen the discussion on this topic in the "Social Touch and Its Implication for Neurologic and Psychiatric Disorders" section. A critical year is 2010, when Morrison and colleagues [22] proposed the hypothesis "skin as a social organ", which gives CT fibers the role of transmission and processing the social dimension of specific touch. Olausson [23] elaborated on the social touch hypothesis during the same period.

### 3. Properties of CT Fibers

Animal and human studies conducted with von Frey monofilaments have established the CT fibers' physiological properties. Their receptive fields are round/oval and consist of one to nine small responsive spots distributed over an area up to 35 mm$^2$ [20], and they respond to innocuous dynamic stimulation across the hairy skin surface [24,25]. The stimulus must be characterized by a force in the range of 0.22–2.5 mN and a velocity of 1–10 cm/s. Velocities lower than 1 cm/s or higher than 10 cm/s lead to a decrease in their firing frequency. This typical response to stimuli has the characteristic upside-down U-firing shape. Once activated, the fibers produce high-frequency trains of action potentials (50–100 impulses/s), with a peak rate of 100 impulses/s. A crucial feature to be considered is that CT fibers are unmyelinated, meaning they have slow conduction velocity (0.6–1.3 m/s) [24]. This aspect cannot be correlated with the discriminative aspect of the touch because we need fast speed in order to discriminate the features of the touched object. Indeed, Aβ fibers are myelinated and have these functions. By contrast, CT fibers are not myelinated.

### 4. Effects of CT Fibers on the Autonomous Nervous System

At the peripheral skin level, it has been demonstrated that CT stimulation evokes a sympathetic skin response. In studies in which skin resistance changes were registered with a constant current electrodermal recording device, using Ag–AgCl electrodes placed on the glabrous skin of the nondominant hand, CT activation was shown to trigger sympathetic arousal [26,27]. Moreover, it has been shown that the stimulation of CT fibers also modulates the level of stress hormones and positively modulates the parasympathetic system. The vagal tone is the activity of the vagus nerve, which is a fundamental component of the parasympathetic branch of the autonomic nervous system. The vagus nerve regulates the automatic systems, and therefore different situations, e.g., anxiety or stress. The vagal tone can be measured by heart rate variability, i.e., the balance between sympathetic and parasympathetic activity in the body in a resting state but also different conditions [28,29].

In humans, it has been demonstrated that among self-stroking, stroking, and being stroked by a partner, just the last condition has a significant autonomic effect, such as the decrement in heart rate [30], even though all of them are stimuli rated as pleasant. In this experiment, the stroke was applied with speed and pressure optimal for the modulation of CT fibers. No visual feedback was provided. Thus, participants could not see each other during the task. An interesting conclusion of the authors is that the absence of the role of the autonomic system during stroking could be due to the absence of visual feedback and because of the absence of CT fibers on the glabrous side of the palm. Moreover, both self-stroking and stroking are perceived as less pleasant than being stroked based on (1) the mechanisms to distinguish self-touch and externally produced touch and (2) the mechanism according to which we correctly predict the pleasantness of the self-touch [30].

A great effect of affective touch is a reduction in pain in healthy subjects [31] and in some diseases, such as chronic pain. Chronic pain is defined as pain for 3 consecutive months even after treatment [32]. Nowadays, the treatment combines pharmacological and physical rehabilitation, but the combination is often insufficient to resolve the problem. New therapies are necessary. Larissa L. Meijer and colleagues [33] reported the effect of affective touch on a 73-year-old woman with a chronic pain diagnosis. Affective touch was applied with a speed of around 3 cm/s, i.e., the optimal speed to modulate CT fibers, and nonaffective touch was applied with a speed of 18 cm/s, outside the range to modulate CT fibers. The patient reported that from the third to the seventh day, the pain disappeared. Importantly, this was not reported during the nonaffective touch treatment.

Another effect of affective touch is the modulation of hormones and neuropeptides. In chimpanzees, grooming with a bond partner increase the urinary secretion of oxytocin [34], supporting the fact that the affective touch can contribute to the production of such neuropeptide. On the other hand, several findings demonstrate that hormones and neuropeptides can regulate the production and perception of the social touch. It is

known that oxytocin is one critical player in affiliative behaviors. Since 1980, research has highlighted the potent effect of this neuropeptide on social interactions in both humans and animals. Drago and colleagues [35] demonstrated that the intracerebroventricular injection of oxytocin induces an increment in grooming in rats. The authors have hypothesized that oxytocin increased affiliative behavior through central mechanisms, in particular through dopamine and opioid systems. Indeed, the opiate receptor antagonist naloxone and the dopamine antagonist haloperidol decreased or suppressed the grooming behavior, respectively. In a study involving female bonobos, oxytocin administration increased grooming behaviors, even if the authors of the study underline that interindividual differences are likely [36]. In humans, the administration of oxytocin may regulate the social evaluations of others after being touched [37]. In the same study, the authors demonstrated an interaction between touch and facial expressions, with angry faces negatively affecting the rating of touch pleasantness [37]. Another study demonstrated that the intranasal application of oxytocin to humans at a resting state induces an increment in heart rate variability, an index of the vagal tone [38].

Finally, another crucial aspect of social touch is the self/other touch distinction since it can be dysfunctional in some neurological diseases. Indeed, differences in the cortical and spinal modulation during self- and other touch have been investigated. Interestingly, researchers have found important differences in the modulation of the insula and the anterior cingulate cortex during these two conditions [39].

## 5. Central Nervous System

Regarding the projections of CT fibers to the brain, it has been demonstrated that they project to the superficial lamina I and inner lamina II of the spinal dorsal horn [18,40,41]. Information reaches the ventromedial posterior thalamic relay nucleus that in turn projects it to the dorsal posterior insular cortex [26,28,42], with a somatotopic organization [43].

Human studies using noninvasive techniques (e.g., functional magnetic resonance imaging and positron emission tomography) have revealed that other brain areas could also be involved in cortical CT processing [37,43–51] for a review:

- The putamen;
- The orbitofrontal cortex;
- The posterosuperior temporal sulcus;
- The medial prefrontal cortex;
- The dorsal–anterior cingulate cortex;
- The pregenual anterior cingulate cortex;
- Superior temporal sulcus.

The identification of the projection pathways involving skin–spinal cord–specific brain areas supports and strengthens the hypothesis that CT fibers belong to the interoceptive system. Therefore, they have a pivotal role in the codification of the emotional component of the touch. In particular, the posterior insula cortex (pIC) has long been thought to have a crucial role in coding social touch. Indeed, studies on patients with a congenitally reduced density of unmyelinated sensory fibers have shown that, in these patients, CT-related touch did not activate the pIC [12].

More specifically, Gordon and colleagues [45] highlighted the existence of a network involved in the codification of CT-related touch (touch on the hairy side), different from that one involved in the codification of not-CT-related touch (touch on the glabrous side). Indeed, in the first case, the touch seems to modulate the pIC–cerebellum–parietal cortex, while in the second case, the involved areas are the pIC–prefrontal cortex–cingulate cortex. Notably, despite both types of touch activating the pIC, only CT-related touch activates the pIC and the connected regions.

## 6. Social Touch and Mirror Neurons

In the last few years, several studies have tried to reveal neural mechanisms for perceiving and understanding social interactions [52]. Understanding the conspecific's

experiences is crucial for social behavior, and according to the mirror neuron theory, this understanding is accomplished by an internal simulation of other's experiences we are observing [53].

Mirror neurons, discovered originally in monkey brains and later in human brains, were first described in a seminal paper in 1992 as a class of monkey premotor cells discharging during action execution and observation [54]. Rozzi and colleagues [55] recorded these neurons also from the cortex of the inferior parietal lobule of macaques. Subsequently, mirror-like neurons were found in different brain areas and animal species (rodents, birds, bats).

After the discovery of mirror neurons for body actions, it has been advanced the hypothesis that an analogous mechanism could be involved in the observation of tactile stimulations. Goldman and Gallese were the first to hypothesize a somatosensory mirror mechanism, which would allow observers to map the observed tactile stimulations onto their somatosensory system [56]. Subsequently, attesting to the social importance of touch, mirror neuron-type responses to observed touch have been reported within the same neural regions activating when the touch is experienced first-hand [57]. Several studies suggest that the posterior portion of the human secondary somatosensory cortex plays a key role in mirror-touch synesthesia [58]. Imaging studies (with functional magnetic resonance imaging) showed that observing another person being touched activates brain areas, such as the primary and secondary somatosensory areas and premotor cortex, which are normally activated when an individual's body is touched [57].

Functionally, the somatosensory cortex has been linked to empathic ability [59], the recognition of emotional expressions [60], and the affective valence and intensity of the observed social touch, such as caressing and slapping someone else's hand. Intriguingly, observing these last two stimuli more strongly activates primary and secondary somatosensory areas than the observation of a simple contact without affective connotation [61].

The insula cortex is also a region of interest in affective mechanisms [62]. Evidence on affective touch in humans suggests that the insular cortex could be an interface between exteroceptive and interoceptive perceptions of social touch during skin-to-skin communication (see Section 7 for all the brain regions involved in the circuit). Morrison et al. [12] showed that the responses to dynamic stroking touch can be "velocity-tuned" and "socially specific". They found that the posterior insula cortex was most activated for CT optimal velocity social stroking rather than non-CT optimal velocities or nonsocial dynamic touch. Evidence supports that individual differences in vicarious responses to touch exist due to personality traits or cognitive state [63,64]. Furthermore, the link between tactile experience and vicarious responding is tangible in patients carrying a heritable mutation, resulting in reduced C-fiber density. This, in turn, implies that these patients not only evaluate the directly experienced CT-related optimal touch as less pleasant [65] but also feel a smaller sense of gratification from observing the same touch compared with control subjects [11].

Empirical evidence shows that the social touch can affect the way we mentally represent our body, and it does this in a somatotopic manner [66]. In addition, the observation of tactile stimulation delivered to a virtual hand is associated with changes in brain activity related to vicarious somatosensation [67]. This last study shows how changes in body ownership can affect the neural activity of brain structures underlying vicarious somatosensation. It is noteworthy that, among those brain regions that changed their neural activity during the vicarious somatosensation, there was also the insula, which not only is the target of CT fibers but is also known to be involved in the processing of somatosensory aspects of altered body ownership [67].

Finally, it seems fair to say that the mirror mechanism allows a translation of the perceived action, regardless of whether we have seen, heard, or felt it, into the identical motor representation of the given action goal, resulting in an embodied link between two individuals [68]. The motor resonance mechanism of mirror neurons most likely represents the neural correlate of understanding others' actions and intentions, which has been described as "embodied simulation" [69–72]. However, actions are not the unique

experiences characterizing interpersonal relationships, which instead imply also sharing affective states, such as emotions and sensations. It is well known that the same brain structures involved in the experience of emotions and sensations are also activated when the same emotions and sensations are recognized in others. Several "mirroring" mechanisms exist in human brains and, thanks to the "intentional consonance" [73], these mechanisms allow us to experience intersubjective relationships and empathize with others.

Overall, several studies have described mirror neurons and mirror-like responses not only during the execution/observation of actions but also for many other kinds of stimuli, such as tactile stimulations, showing that merely viewing touch involves the observers' somatosensory cortices. There is a large body of evidence that indicates that information about others' actions, emotions, sensations, and communicative messages are mapped onto the same beholder's neural substrates devoted to those first-person processes [74]. Therefore, it would appear that the mirror mechanism allows for a basic widespread remapping of other-related information belonging to a large variety of domains (especially social cognition and social behavior) onto primarily self-related brain structures. Notably, a few of these structures, which are involved in the processing of emotions, belong to the mirror network, and they are also targets of CT fibers.

In the next section, we will describe the brain structures underlying the mirror mechanism and explore how this network can be embedded into a larger circuit to explain the whole social behavior in typical and atypical individuals.

## 7. From Social Brain to Social Behavior

Neuroanatomical studies have been performed on areas where neurons with mirror properties have been found. Through those studies, we can identify two different mirror neuron networks: sensorimotor and emotional (Figure 1A). The sensorimotor network [75–77] (Figure 1A, purple) includes, in addition to the ventral premotor cortex [78], the primary motor cortex [79], inferior parietal lobule [80], the anterior intraparietal area [75,81], the dorsal premotor [82,83] and mesial premotor [84–86] cortices, the prefrontal cortex [87], and the secondary somatosensory cortex [88]. According to additional evidence in humans, the basal ganglia and the cerebellum might also have a role in these networks [89]. The emotional network (Figure 1A, green) includes the anterior cingulate cortex, the amygdala [90], and the insula [91,92]. Notably, the insular and secondary somatosensory cortices are the common hubs of these two networks, suggesting that they could represent a bridge between action/perception and emotion. Furthermore, the network underlying affective touch (Figure 1B) includes the thalamus, the primary somatosensory cortex, the insular cortex, the medial prefrontal cortex, and the amygdala. Interestingly, the insular cortex is the cortical node of both mirror neurons and affective touch systems. Additionally, it is known that, in neurotypical individuals, CT optimal touch activates social brain networks, and they include, as we already underlined, the bilateral insula together with the parietal operculum, the orbitofrontal cortex, the primary somatosensory cortex, the posterior superior temporal sulcus, and the inferior frontal gyrus [93–96].

To conclude, these studies suggest that, in individuals with neurotypical development, a central hub exists that plays a crucial role in the processing of emotional and social stimuli and is also involved in the processing of affective touch. Dysfunctions within one of the networks can have severe consequences, and, in a few cases, they can also impact the functioning of other networks. Nonetheless, this link is also remarkable since it can offer new insight into treatment options for the diseases described below. We will extensively discuss this topic in subsequent sections.

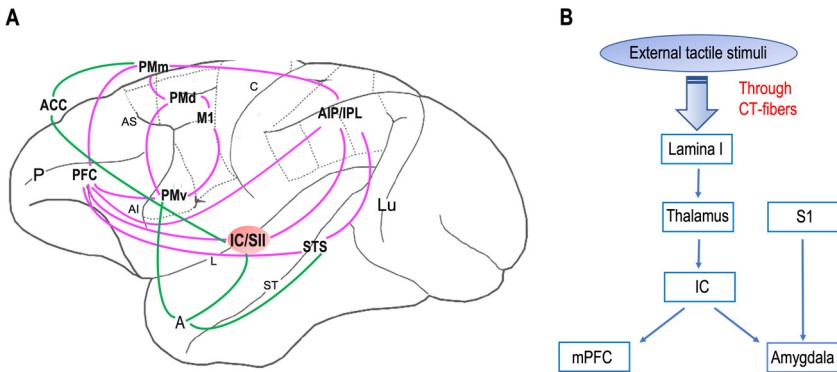

**Figure 1.** Neural networks of mirror neuron system and affective touch: (**A**) Organization of primate sensorimotor (purple) and emotional (green) mirror neuron networks based on macaque neuroanatomical studies on areas in which neurons with mirror properties have been found. The sensorimotor network includes, beyond the ventral premotor cortex (PMv) and the inferior parietal lobule (IPL), the primary motor cortex (M1), the dorsal premotor cortex (PMd), the mesial premotor cortex (PMm), the prefrontal cortex (PFC), the anterior intraparietal area (AIP), and the secondary somatosensory cortex (SII). The emotional network includes the anterior cingulate cortex (ACC), amygdala (A), and insula (IC). Abbreviations: P, principal sulcus; AI, arcuate inferior sulcus; arcuate superior sulcus; L, lateral sulcus; C, central sulcus; ST, superior temporal sulcus; Lu, lunatu. (**B**) The circuit of affective touch includes external tactile stimuli, which produce mechanical energy. This mechanical energy, received at the somatosensory nerve endings in the skin, is transduced and transmitted via CT fibers up through lamina I and to thalamic nuclei (i.e., ventromedial posterior nucleus, ventral posterior inferior nucleus) and processed at the insula (IC), before reaching the social centers in the cortex (e.g., medial prefrontal cortex, mPFC) and amygdala. Abbreviations: S1, primary somatosensory cortex.

## 8. Social Touch and Its Implication for Neurologic and Psychiatric Disorders

A strong body of evidence supports the finding that social touch is relevant to several neurological and psychiatric disorders since CT fibers' dysfunction appears involved in these pathologies. For example, it has been demonstrated that, in patients displaying a congenitally reduced density of unmyelinated sensory fibers, the pIC does not activate when a slow and gentle touch is applied. This is most likely due to poor CT afferent inputs to the cortex [12].

Patients carrying a rare sensory neuropathy (a rare disorder of nerve cell bodies of the large primary sensory neurons) lack myelinated Aβ fibers on their skin in extended skin areas. However, they have intact unmyelinated CT fibers. These types of patients have been well described in the literature [97,98]. In particular, the motor functions of these patients have been investigated for the proprioceptive deficit that characterizes them. Briefly, these patients could be potentially able to detect the stimulation of unmyelinated low-threshold mechanoafferents. Indeed, in a two-alternative forced choice test, they detected light touch applied to the forearm skin, an area rich in CT fibers, but could not detect the same type of stimuli when applied to the glabrous skin of the hand, where CT fibers are poor. They described the sensation of the stimulus applied as weak, with pleasant connotations and no pain or itch. Their ability to localize CT stimulation was poor. They also had difficulties detecting 50 Hz vibratory stimuli, which are known to activate Aβ afferents but not CT fibers. Finally, fMRI studies highlighted that selective CT stimulation activates the posterior insular cortex and deactivated both primary and secondary somatosensory cortices [25–28,44].

In the literature, a hereditary disorder has been described that is associated with a nerve growth factor beta gene mutation, which in turn causes the denervation of unmyelinated skin afferents. Even if it does not cause a complete loss of nerve growth factor function [99], the mutation can lead to a severe-to-moderate reduction in unmyelinated C fibers, together with a moderate reduction in thinly myelinated Aδ fibers, excluding any

other neurological or cognitive abnormalities [100]. Patients with the hereditary sensory and autonomic neuropathy type V mutation are a well-defined population of consanguineous individuals geographically scattered in the Norrbotten region in the north of Sweden, along the Tornea River Valley. When compared with healthy control subjects, these patients had lower pleasantness ratings, and their rating patterns across different velocities (0.3–30 cm/s) deviated from normal rating patterns. Moreover, neuroimaging studies showed that, in these patients, the stroking speed of 3 cm/s did not activate the insular cortex, and there was no difference in the modulation of the insular cortex between the optimal and nonoptimal velocities of CT activation (3 cm/s and 30 cm/s, respectively) [101].

Anorexia nervosa is an eating disorder characterized by abnormally low body weight, an intense fear of gaining weight, and a distorted perception of the body (overestimation of body size and shape). People with anorexia place a high value on controlling their weight and shape, using extreme efforts that tend to significantly interfere with their lives. Importantly, these patients have deficits in interoceptive processing and their internal physiological condition interpretation, creating and maintaining a mismatch between internal and external bodily sensations. Some studies have reported that anorexia nervosa patients perceive touch that activates CT fibers as less pleasant than control healthy individuals [102]. Nevertheless, there is no difference between anorexia nervosa and healthy people concerning the brain regions involved in pleasant touch coding. Indeed, the activation of the posterior insula appears to be present in patients with anorexia nervosa. However, there is no agreement in this field regarding neuroimaging results. Recently, it has been proposed that the differences in touch perception could be due to alexithymia (the absence of emotional awareness) [102]. An interesting aspect that has not yet been fully understood and investigated is the perception of viewing others being touched with a pleasant touch. In healthy people, the view of others being touched activates the same brain areas modulated by being touched oneself [57].

Croy and colleagues [102] investigated this topic in patients with one or more psychiatric disorders (somatoform disorders, post-traumatic stress disorders, anxiety, somatoform disorders, and mood and affective disorders). Patients reported significantly fewer interpersonal touch episodes compared with a healthy control group, regardless of whether they lived alone or not. Moreover, patients differed from the control group in terms of touch pleasantness but not regarding its awareness. Interestingly, the diagnosis correlated with the touch pleasantness rating but not gender, age, and medications used. In the same study, the authors also tested touch perception in subjects with autistic traits. They compared patients and healthy groups and pointed out that, in both groups, autistic traits were related to affective touch awareness. The authors also concluded that the degree of affective touch awareness could be a diagnostic criterion for autistic traits.

Table 1 summarizes the disorders in which the dysfunction of CT fibers seems to be involved, which are reviewed in this study.

Finally, there are a few clinical conditions (e.g., autism spectrum disorder) related not only to CT fibers but also to the mirror neuron system. As highlighted in the previous section, extensive effort has been made to describe both mirror neurons and the affective touch system. However, more important than defining a neural network for the mirror mechanism or social touch is an understanding of whether and how the nodes of these networks interact with each other, and what the functional implications are of both typical and aberrant interactions at different stages of development. Indeed, individuals with a diagnosis of autism spectrum disorder display different levels of functional connectivity across all these networks during pleasant touch [95,96,103], as well as the hypoconnectivity of the right primary somatosensory cortex and the right superior temporal gyrus during innocuous tactile stimulation [95]. These findings support observations of reduced pleasantness discrimination between CT-related optimal touch and non-CT optimal touch [104] and the aberrant processing of non-CT optimal touch in autism spectrum disorder [105]. However, given the heterogeneity of autism spectrum disorder, some studies report no variations in affective and discriminative processing between individuals with autism

spectrum disorder and age-matched controls [103,104]. Contrarily, during the observation of socially/emotionally salient stimuli, individuals with autism spectrum disorder demonstrated atypical hypoactivation of the inferior frontal gyrus [106], which belongs to the human mirror neuron system and is involved in the processing of the emotional aspects of vicarious touch [107]. Additionally, individuals with autism spectrum disorder showed a decreased functional connectivity between the somatosensory cortex and the superior temporal gyrus, both involved in affective empathy, but typical activity in the temporoparietal junction [95], when they observed positive social touch (i.e., hugging, and caressing). Altogether, these findings, and the data available supporting the fact that subjects with autism spectrum disorder have a typical theory of mind [108], suggest that the social impairments displayed by these people are more related to the inability to "experience" the socioemotional attributes of affective touch rather than the inability to "interpret" the experiences of others [109].

**Table 1.** Review of disorders involving the CT-fiber system.

| Disorder | Causes | Affected Fibers | Perception of Touch CT Related | Perception of Touch Not-CT Related |
|---|---|---|---|---|
| Anorexia nervosa | Multifactorial | | Less pleasant than control | |
| Rare sensory neuropaty | After infection (e.g., mononucleosis) | Absence of myelinated Aβ fibers on their large skin areas. Nevertheless, they have intact unmyelinated CT fibers | The ability to spatially localize CT stimulation is very poor | No perception |
| Disorder associated with a nerve growth factor beta (NGFB) | Gene mutation that determines the loss of NGFB function | Severe to moderate reduction of unmyelinated C fibers and a moderate reduction of thinly myelinated Aδ fibers | The pleasantness ratings of these patients were lower and the rating pattern across the different velocities (0.3–30 cm/s) deviated from the typical and normal rating pattern, therefore to the typical inverted U-shaped curve, correlated with CT discharge across velocities | |
| Psychiatric disorder | Multifactorial | | Less pleasant than healthy subjects | |
| Autism spectrum | Multifactorial | | Aberrant behaviour (defensiveness); Reduction of the central network involved in CT related | Impaired; Increase of the central network involved in discriminative touch |

## 9. Touch as Therapy?

Human studies have shown that affiliative touch and massage could be useful in several clinical pictures. Recently, a meta-analysis on the effect of therapeutic massage on symptoms of Parkinson's disease provided evidence that therapeutic massage could alleviate the typical motor symptoms of this illness, improving motor functions even though it did not ameliorate the quality of daily life of patients in comparison to healthy subjects [110]. In the reviewed studies in this meta-analysis, the applied massages were traditional Chinese Tuina, a common massage technique targeting the limbs of patients; acupressure; and Thai massage.

Billhult and colleagues [111] highlighted the positive effects of touch on women with breast cancer. Indeed, they demonstrated a decrease in heart rate and systolic blood pressure. A reduction in heart rate was also reported in psychiatric patients (e.g., mood disorders or borderline personality disorder) during massage therapy, concomitant with a

decrease in cortisol levels and improvements in depression, hostility, and aggression [112]. It has been demonstrated that massage is also effective in improving the quality of life of multiple sclerosis patients [113]. Moreover, findings provide evidence supporting the importance of oxytocin for enhancing positive behavioral and neural responses to social touch in the form of manually administered massage. These results suggest that a combination of intranasal oxytocin and massage may have therapeutic potential in autism [114].

Recently, Tsuji and colleagues analyzed the salivary oxytocin concentration in children with a diagnosis of autism spectrum disorder when receiving a massage from their mothers [115]. The salivary concentration of oxytocin was higher during the massage than before in both mothers and children. This study is interesting since people with autism spectrum disorder usually avoid physical contact, but in this study, the touch was predictable. Notably, this kind of predictable contact decreases the aversion to touch. Finally, mothers reported improvement in social interactions and that they seem more relaxed than usual.

Lately, mechanical affective touch therapy has been proposed as a useful treatment for controlling symptoms of anxiety disorders. The applied stimulation through mechanical affective touch therapy has CT fibers as its target. Electroencephalography studies revealed that the treatment also led to increased occipital theta and alpha oscillatory activity [116]. Interestingly, alpha oscillatory activity seems to be correlated with interoceptive awareness and a relaxing state, whereas an increase in theta oscillation is associated with a mindful state. Finally, it has been demonstrated that CT-related touch could help to alleviate chronic pain. For example, Baumgart and colleagues [117] recently reported that psycho-regulatory massage therapy in patients with chronic back pain has a positive effect in terms of a decrement in pain.

Preterm newborns represent another patient population for whom CT-related touch is increasingly used as a treatment. There is evidence that tactile stimulation improves the immune system in premature newborns [118]. Additionally, it is known that maternal care is crucial for brain development, and deficits in maternal care can result in maladaptive behaviors in adulthood. The results of a recent study [119] suggest that maternal care affects serotonergic neural activity during early life, and the study provides key insights into how maternal care affects the adaptive/maladaptive development of brain circuits implicated in adult pathology. Thus, skin-to-skin care has become common in several hospitals (e.g., Scandinavian Neonatal Intensive Care Units since the 1980s) [120]. The International Workshop on Kangaroo Mother Care, 2009, recommends the implementation of continuous kangaroo mother care as the gold standard pervading all medical and nursing care, based on empirical studies and clinical guidelines. Additionally, they suggest that kangaroo mother care may be used during terminal care in agreement with parents [120]. However, skin-to-skin care and touch therapy at the end of life have been under-researched. Future investigations in these areas are needed. Nonetheless, a growing body of evidence supports the notion that physical contact, gentle stroking, cuddles, and tender loving care are of central importance for a dying baby [121]. Given the importance of maternal contact during early life, several hospitals have launched volunteer cuddling programs for all infants admitted into the neonatal intensive care unit. These programs usually utilize trained volunteers to cuddle infants when caregivers are not available. At St. Michael's Hospital, Toronto, a pilot cohort study was performed with a retrospective control group to assess the impact of volunteer cuddle programs on the length of stay and the feasibility of implementing volunteer cuddling programs for infants with neonatal abstinence syndrome [122]. The length of stay was used as a surrogate marker to measure the impact of cuddling on these infants. The study results suggest that volunteer cuddling programs may reduce the length of stay in infants with neonatal abstinence syndrome. However, larger prospective cohort studies are needed to confirm these results.

Altogether, these studies emphasize how several touch techniques (i.e., psychoactive massage therapy) have been introduced into different fields, from the treatment of psychiatric disorders (depression, anxiety, or psychosomatic disorders) to the treatment

of chronic pain, such as in patients with cancer [123]. Understandably, patients gladly accept this kind of treatment, compared with the more invasive ones. However, it faces skepticism or resistance among clinicians and health professionals for several reasons [124]. First, massage and touch-based therapies represent a heterogeneous group of interventions lacking uniformity in terminology and definitions across studies [125]. Indeed, depending on the workgroup, touch techniques can include massage, affective touch, affiliative touch, etc. Second, it can be argued that many studies have reported the effect of massage therapies but not the effect of massage therapies targeting CT fibers [126], contributing to the variability in the field. Third, limited evidence is available from prospective clinical trials that test the touch therapy efficacy [125]. Moreover, when findings are available, they are often of low quality, mainly because of inaccurate or vague definitions of interventions and difficulties in designing control settings.

That said, we believe there are different motivations to encourage future investigations in academic medicine, clinical psychology, and basic research to take steps to make touch therapy attractive and accepted as an affective-oriented assessment and intervention, resulting in a more precise and standardized classification of clinical conditions affecting emotional processing. Furthermore, the recent pandemic has highlighted the importance of interpersonal touch by showing the psychological impact of the social distancing measures enforced by authorities. Finally, a growing body of literature on clinical studies strongly supports the effectiveness of various types of massage despite various methodical deficits and the heterogeneity of the studies performed to date, to the best of our knowledge [124]. For all these reasons, we need valid and high-quality scientific studies beyond methodologically rigorous clinical trials to evaluate and confirm the beneficial effects of touch therapies. Further empirical research elucidating the mechanisms underlying the effects of massage therapy on the brain is also needed to extend our knowledge about key concepts of cognitive neuroscience and neuropsychology and to strengthen the acceptance of this kind of intervention by medical professionals and the scientific community.

We hope this review will spark discussion among the groups studying these concepts to reach some common ground on definitions, methodology, and approaches.

## 10. Conclusions

Because of the critical role of social touch in human well-being and the evidence supporting its use as a treatment for both newborns and adults, more detailed studies are necessary. Indeed, investigating the correlation between each disorder and the corresponding CT dysfunction and/or distorted perception of affiliative touch should help to develop new treatments based on massage and CT-fiber stimulation in association with pharmacological therapies.

Another issue that remains to be delved deeper into is the role of the mirror network in neurological and psychiatric disorders. It is still unclear whether and to what extent the mirror circuit is involved in specific disorders in which patients have a touch perception significantly different from healthy people. Thus far, shared neural networks underlying the process of another individual's action, emotion, or sensation have been identified. The vicarious activation of brain networks can be an automatic and unconscious process or overt experience of the emotion or sensation observed in another person. Studying these vicarious processes, together with interpersonal properties such as empathy, allows us to better understand the relationship between the social brain and social behavior. The activation of an emotional brain, when we observe or recognize others' emotions, helps us to synchronize our behaviors with others during social interactions. Importantly, this field has significant implications in identifying new therapies for people with psychiatric or neurological diseases characterized by social impairments such as autism, depression, or schizophrenia. The knowledge of the neurobiology of social processes will help to generate knowledge for targeted and appropriate treatments.

Ultimately, it would be useful to develop new therapies based on the protocols targeting CT fibers in order to alleviate symptoms in several clinical conditions. These treatments

should improve not only behavioral symptoms, as seen in autistic children, but also the autonomic state. Importantly, new investigations might underline the possible effect of CT stimulation in pathological conditions in terms of circuit modulation in the "social brain".

**Funding:** This research received no external funding.

**Institutional Review Board Statement:** Not applicable.

**Informed Consent Statement:** Not applicable.

**Data Availability Statement:** Not applicable.

**Conflicts of Interest:** The authors declare no conflict of interest.

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
