# Peer review of "Social Touch: Its Mirror-like Responses and Implications in Neurological and Psychiatric Diseases"

_neurosci, doi:10.3390/neurosci4020012_

Round 1

Reviewer 1 Report

In the paper “Social touch: its mirror-like responses and implication in neurological and psychiatric disease” the authors reviewed the available evidence on the neural basis and possible implications of social touch. To this aim the authors focused the role of CT fibers in the autonomous and central nervous systems, with a particular emphasis on their implication in mirror-like perceptions. The timely and sound content should be sufficient to warrant publication. Nevertheless, the manuscript might benefit from a more mechanistic explanation of why/how CT fibers could be linked to emotional processing.

1) There might be room to better highlight the role of social touch in the mental representation of one’s own body and mirror-like responses such as embodiment. For example, it could be worth noting that social touch affects the way we mentally represent our body and in a very somatotopic manner (Martinez VR et al 2022 Neuroscience) and that the observation of tactile stimulation delivered to a virtual hand is associated with changes in the brain activity related to vicarious somatostimulation (Pamplona GSP et al 2022 Cerebral Cortex). On this basis, it could be proposed that the role played by CT fibers in the somatosensory-depending cognitive aspects related to body representation and embodiment might constitute the basis for accurate touch-related emotional processing, and that distorted somatosensory processing due to dysfunctional CT fibers might results in aberrant body representation and/or embodiment and, therefore, impaired emotional responses. This perspective might offer a stronger background and/or a more mechanistic explanation of one the possible reasons why CT fibers are so closely linked to emotional processing and mirror-like responses.

2) It could be worth discussing how the present study could help the development of affective-oriented experiments in humans and, therefore, contribute to extend the knowledge about key concepts of cognitive neuroscience and neuropsychology. For example, it might be proposed that the implementation of experimental protocols targeting CT fibers in future affective-oriented assessments and interventions could result in a more precise/standardized classification of clinical conditions affecting emotional processing, which in turn could provide a better focus on the cognitive neuroscience concepts (e.g. empathy, body ownership, vicariousness, etc) addressed by the specific programs. This might offer a stronger impact of the present paper in a broader framework. What do the authors think?

MINOR POINTS

3) Abstract – It might be informative to mention the nature of the present review paper (systematic review, literature review, etc).

4) Abstract – It seems that “more-over” should be “moreover”.

5) Acronyms – There are a lot of acronyms that are used only a very few times (e.g. AIP, among others) or even only once (e.g. SOMSoM, among others). In order to help readability, it is suggested to keep acronyms only for “labels” that are effectively repeated many times (e.g. CT). After this selection, if the number of remaining acronyms will be still high, it could be useful to add something like a “List of Abbreviations” at the beginning of the paper. This approach should not decrease the clarity of Figure 1, where adding a legend to explain the acronyms could be even more clear than being forced to read the figure caption in order to understand what each abbreviation means.

Author Response

We thank the reviewer for the important suggestions and comments.

We modify the manuscript following them and we think now the manuscript is improved.

Reviewer 2 Report

In this review, the authors survey the literature on CT-related social touch, focusing on its neural circuitry, its relationship to the mirror system, and its implications for neurological and psychiatric disorders. Overall, it is a thorough and timely summary of this interesting field and has the potential to be a valuable addition to the literature.

Here are some suggestions to help strengthen the paper.

Major:

As the title suggests, there are two main parts to this review, the CT-related mirror system (sections 6 and 7) and the CT-related diseases (section 8), but it is unclear how these two topics relate to each other. They seem rather disjointed. For example, none of the diseases discussed in section 8 are related to the mirror system.

The only exception is the second half of the first paragraph of section 7, where the authors discuss ASC, which is both relevant to the mirror system AND a disease. Therefore, I would recommend moving the content between lines 247 to 290 somewhere in section 8 or after section 8, highlighting the discussion of CT AND mirror system related disorders.

Accordingly, I suggest that the discussion of CT and mirror network related disorders be expanded in Section 10. (In fact, Lines 272 - 290 seem more appropriate as a discussion section)

Minor:

  1. Line 19: Why were premature infants considered "nonphysiological"?
  2. Line 20 and line 374: the meaning of "resume" is confusing
  3. Line 58: this paragraph appears to be unfinished?
  4. Line 85: "We will ..." is duplicated to the content in parentheses in line 84.
  5. Line 101: Why is the slow speed correlated with social aspects?
  6. Line 110: needs a citation
  7. Line 135: this paragraph is about how oxytocin regulates the perception of social touch, not the opposite (line 122: effect of affective touch on oxytocin). Also, the next paragraph seems irrelevant.
  8. Line 168: Include the full name of pIC at its first occurrence.
  9. Line 205: It is better to start a new paragraph with "The insula cortex...".
  10. Line 208: What does "paragraph 5" refer to here?
  11. Table 1: The fourth column is too crowded. Consider adding horizontal lines to the table.
  12. Some words have unnecessary hyphens. For example, line 12 "more-over", line 18 "pro-vides".

Author Response

We thank the reviewer for the comments. We modified the manuscript followed the suggestions in order to improve the manuscript.

Reviewer 3 Report

I would like to thank the Authors and Editors for the opportunity to review this manuscript. I believe the paper is relevant as research on social or affective touch is increasing, and it plays an important role in how people interact with each other. Research on social touch and its relationship to mirror neurons can provide insight into how touch can be used as a tool in the future to improve well-being in psychiatric diseases.

I believe the overall quality of the manuscript is good, and the literature reviewed is appropriate. Although I have some minor comments and recommendations which I think will improve the Authors work.

1. The first paragraph of the introduction has no citations, is this correct, are these the Authors statements?

2. Statement starting at page 2, line 55 seems to be missing a citation.

3. Paragraph 2, on page 2, seems to only have one sentence. I recommend incorporating this sentence into another paragraph.

4. I would like to recommend to the Authors to also incorporate literature on self- other-touch distinction, especially after third paragraph in page 3. There is a PNAS paper on the matter.

5. Towards the end of paragraph 3, on page 3, Authors discuss about pleasantness of touch. It might be interesting to discuss pain here, and its relationship to touch.

6. On line 122 on page 3 there is another paragraph of one sentence, again I recommend the incorporation to another paragraph.

7. Authors talk about relevant changes in vagal tone; however, I believe a definition of this concept would be appreciated by the readers.

8. In line 145 in page 3 the Authors state: “… ultimately depend on the motivation of the subject”. Please describe in which way to be more precise.

9. I believe there is a typo in line 190, page 3: rodents, birds, buts?

10. Within the heading 7. From social brain to social behaviors, Authors discuss networks related to mirroring behaviors, and after start discussing autism spectrum condition touch alterations. I believe there is a bit of a jump here, readers could benefit from more content on touch behaviors in neurotypical development beforehand.

 11. Autism spectrum condition is mentioned again after abbreviation has been introduced, please correct.

12. In line 382, I believe there is a dot that should be a comma.

Author Response

We thank the reviewer for the comments. We modified the manuscript followed its suggestions and we think now the manuscript is improved in comparison to the first version.

Round 2

Reviewer 1 Report

accept